# Effect of Drought, Nitrogen Fertilization, Temperature and Photoperiodicity on Quinoa Plant Growth and Development in the Sahel

**Jorge Alvar-Beltrán [1],\*** , **Abdalla Dao [2]** , **Anna Dalla Marta [1]** , **Coulibaly Saturnin [2]** , **Paolo Casini [1]** , **Jacob Sanou [2]** and **Simone Orlandini [1]**

[1] Department of Agriculture, Food, Environment and Forestry (DAGRI)-University of Florence, 50144 Florence, Italy; anna.dallamarta@unifi.it (A.D.M.); paolo.casini@unifi.it (P.C.); simone.orlandini@unifi.it (S.O.)

[2] Institut de l'Environnement et de Recherches Agricoles (INERA), Bobo Dioulasso BP910, Burkina Faso; dao_abdalla@yahoo.fr (A.D.); saturnincoulibaly@gmail.com (C.S.); jsanou24@yahoo.fr (J.S.)

\* Correspondence: jorge.alvar@unifi.it; Tel.: +39-275-5741

**Abstract:** Drought, heat stress, and unfavorable soil conditions are key abiotic factors affecting quinoa's growth and development. The aim of this research was to examine the effect of progressive drought and N-fertilization reduction on short-cycle varieties of quinoa (c.v. *Titicaca*) for different sowing dates during the dry season (from October to December). A two-year experimentation (2017–2018 and 2018–2019) was carried out in Burkina Faso with four levels of irrigation (full irrigation—FI, progressive drought—PD, deficit irrigation—DI and extreme deficit irrigation—EDI) and four levels of N-fertilization (100, 50, 25, and 0 kg N ha$^{-1}$). Plant phenology and development, just like crop outputs in the form of yield, biomass, and quality of the seeds were evaluated for different sowing dates having different temperature ranges and photoperiodicity. Crop water productivity (CWP) function was used for examining plant's water use efficiency under drought stress conditions. Emerging findings have shown that CWP was highest under DI and PD (0.683 and 0.576 kg m$^{-3}$, respectively), while highest yields were observed in 2017–2018 under PD and its interaction with 25 to 50 kg N ha$^{-1}$ (1356 and 1110 kg ha$^{-1}$, respectively). Mean temperatures close to 25 °C were suitable for optimal plant growth, while extreme temperatures at anthesis limited the production of grains. Small changes in photoperiodicity from different sowing dates were not critical for plant growth.

**Keywords:** Burkina Faso; *Chenopodium quinoa* Willd.; heat stress; irrigation; crop water productivity

## 1. Introduction

Sustainable irrigation strategies that can increase crop water productivity are of growing interest in arid and semi-arid regions [1]. This is the case of the Sahel, with a rapid growing population and an agricultural system extremely dependent on weather; 98% of its agriculture is rainfed, in addition to experiencing larger inter/intra annual rainfall variability [2,3]. In Burkina Faso, one-third of the country's surface area is degraded, with an estimated annual expansion of degraded land of 3600 km$^2$ [4]. In this region, changes in weather patterns are expected to have a negative impact on the yields of major cereals [5,6]. Some studies conducted in Burkina Faso estimate a yield reduction of 10% for sorghum, close to 17% for maize, and up to 23% for millet by 2050, and is expected to worsen with changing climatic conditions [7–9]; all of the previous crops having a C4 photosynthetic pathway. Even though C4 crops have the most efficient form of photosynthesis [10], crop's with C3 photosynthetic pathways can benefit more from increasing $CO_2$ concentrations [11,12]. This is because

optimal $CO_2$ atmospheric concentrations for enhancing the photosynthetic rate of C3 crops have not yet been reached [11,12].

Quinoa is an herbaceous crop belonging to the C3 group of plants, besides of having a high nutritional value [13,14]. In the last decades, there has been an intensification of scientific research, and in 2013, the International Year of Quinoa was declared—all of which, doubling the number of countries growing quinoa since then [15]. In the Sahel region, several countries (Senegal, Mauritania, Mali, Burkina Faso, Niger, Chad, and Sudan) are now growing quinoa for research purposes, as well as seeking to promote quinoa for local consumption [15]. The latest scientific efforts have focused on quinoa's adaptability under adverse environmental conditions. Its resilience to abiotic stresses is the result of a wide genetic variability [16], giving the plant an excellent salt-tolerance, up to 40 dS m$^{-1}$, and drought-resistance, with water requirements as low as 230 mm for c.v. *Titicaca* [17–20]. The crop water productivity (CWP; amount of biomass in kg per volume of water supply in m$^3$) of quinoa is low, and can be increased: if water losses from evaporation are reduced, if the negative effect of drought stresses at specific phenological phases are avoided, and if unfavorable conditions during crop growth, (i.e., pests and diseases) are diminished [21]. In addition, quinoa can adapt to most types of soils, but differences in the literature arise when assessing crop nitrogen-uptake. Some studies argue that maximum yields are reached at 80 kg N ha$^{-1}$, but yields can differ according to the soil type, being highest among sandy–clay–loam soils [22,23]. On the contrary, some authors have acknowledged that increasing N-fertilization does not determine quinoa growth nor yield [24].

In this paper, the effect of drought and nitrogen fertilization on plant's phenology and physiological responses of short-cycle varieties is evaluated. A crop's performance during the dry season in the Soudanian belt and its adaptability to unfavorable soil and climatic conditions was monitored. Moreover, different sowing dates were tested and some agricultural planning practices for farmers were proposed. Finally, as a climate resilient and C3 crop, this research aims to look beyond the traditional grown C4 cereals (e.g., maize, sorghum, and millet) and has selected quinoa for enhancing food security and nutrition in Burkina Faso under changing climatic conditions.

## 2. Materials and Methods

### 2.1. Experimental Setup

This experiment was carried out between 2017 and 2019 at Institut de l'Environnement et Recherches Agricoles (INERA) Farako-Bâ research station (11°05' N; 4°20' W), Burkina Faso. The study area was characterized for having a tropical savannah-wet and hot climate, with a well-defined dry season (from November to April). In the first year, 2017–2018, the experimental design was organized in a split-block design with a multiple factor analysis of variance (ANOVA), with three levels of irrigation according to the potential evapotranspiration (PET) (FI 100% PET, progressive drought (PD) 80% PET, and deficit irrigation (DI) 60% PET) and three levels of N-fertilization (100, 50, and 25 kg N ha$^{-1}$), while each treatment having three replicates (Figure 1). For the second year, 2018–2019, the experimental set-up was as follows: two levels of irrigation (FI 100% PET and extreme deficit irrigation (EDI) 50% PET), two levels of N-fertilization (100 and 0 kg N ha$^{-1}$), with each treatment having four replicates (Figure 1). Between 2017–2018 and 2018–2019 slightly different irrigation levels and N-fertilization levels were used in order to identify and minimized the gaps between inputs (crop N-fertilization and irrigation requirements) and outputs (losses in the form of volatilization, leaching and water-surface runoff).

Throughout the two years of experimentation, 2017–2018 and 2018–2019, maize was the crop grown during the rainy season whereas quinoa during the dry season. Moreover, for the first year of experimentation, quinoa (c.v. *Titicaca*) was sown on the 4th November (hereinafter, 4-Nov.) and 8th December (hereinafter, 8-Dec.); whereas for the second year, the sowing date was the 25th October (hereinafter, 25-Oct.) and 19th November (hereinafter, 19-Nov.). The sowing rate was 10 kg of seeds per hectare, with 50 cm distance between rows and 10 cm between plants, with a total density of

200,000 plants ha$^{-1}$. Prior to the sowing of quinoa, the field was amended with 5 t ha$^{-1}$ of compost (50.2% organic matter) as well as phosphate (26.7% $P_2O_5$) at a rate of 400 kg P ha$^{-1}$. N-fertilization was applied twice, 2 to 3 weeks and 4 to 5 weeks after sowing, at a same concentration rate.

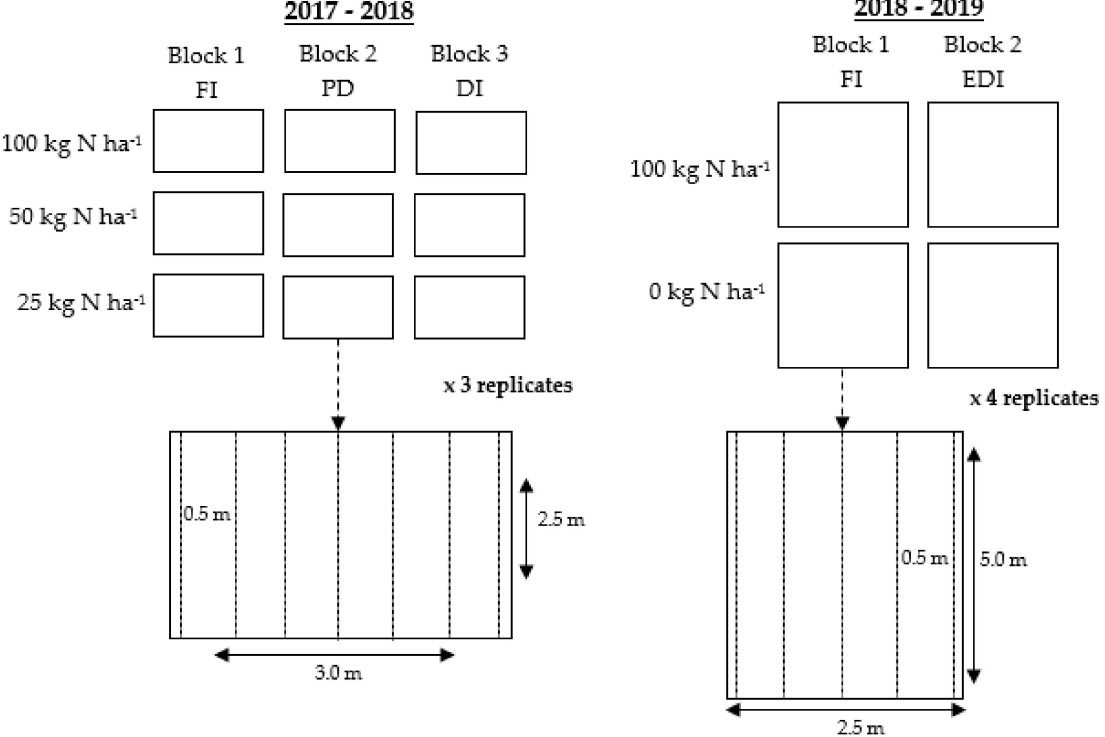

**Figure 1.** Experimental field design with different levels of irrigation (full irrigation (FI), progressive drought (PD), deficit irrigation (DI), extreme deficit irrigation (EDI)), and N-fertilization (100, 50, 25 and 0 kg N ha$^{-1}$) for the two year-experimentation (2017–2018 and 2018–2019).

## 2.2. Sampling and Measurement

Maximum and minimum air temperatures, maximum and minimum relative humidity, and daily rainfall values were obtained from an agro-meteorological station placed nearby to the experimental field. Irrigation was measured using a water-counter placed at the entry of each irrigation block and was drip-irrigated at a rate of 1.05 L h$^{-1}$. Crop's daily potential evapotranspiration (PET$_c$) was calculated using the following formula [25]:

$$ET_o = 0.023 \ (T \ mean + 17.78) \ R_o \ (T \ max - T \ min)^{0.5} \tag{1}$$

where: $R_o$ = solar radiation (MJ m$^{-2}$ day$^{-1}$) at a given month and latitude [26]; *T mean* = mean daily temperature (°C); *T max* = daily maximum temperature (°C); *T min* = daily minimum temperature (°C).

The different phenological phases were measured and analyzed separately according to the different sowing dates, being the following: emergence (E), two leaves (2L), four leaves (4L), eight leaves (8L), panicle formation (PF), flowering (FL), and milky grains (MG) at 5, 15, 18, 24, 30, 40, and 60 days after sowing (DAS), respectively with a total of 100 samples per plot. The phenological phase was determined once 50% of the plants had reached a given stage. Regarding plant's morphology, the plants' height (10 measurements per plot), number of branches (10 per plot), panicle length (10 per plot), stem diameter (5 per plot), root depth and horizontal length (1 per plot) were all measured at physiological maturity. In addition, the plants' performance was measured for the following crop parameters: grain yield per plant (12 measurements per plot) and thousand grain weight (3 per plot). The crop water productivity (CWP) was defined as the ratio between biomass production and water

applied (kg m$^{-3}$). The harvest index (HI) was calculated as the ratio between yield and biomass, as a percentage. For the second year, the canopy cover was measured throughout the growing cycle using a mobile-application, Canopeo, developed by the Oklahoma University [27]. Measurements were taken at 60 cm distance from the top of the canopy with an image cover of $75 \times 50$ cm. During the 2018–2019 experimentation, the canopy cover was measured six times: at 24, 34, 40, 49, 70, and 85 DAS for the sowing on the 25-Oct., and at 24, 34, 40, 60, 75, and 84 DAS for the sowing on the 19-Nov. The calculation of the cumulative growing degree-days (CGDD in °C) was done using the following formula, and where *Tbase* was equal to 3 [28]:

$$GDD = \frac{(Tmax + Tmin)}{2} - Tbase \tag{2}$$

Finally, prior to sowing, five soil samples were extracted from each experimental field at 0–20 cm and 20–40 cm depth. The main physic-chemical properties of the soil, texture, pH, N, P, K, C, and organic matter were analyzed.

*2.3. Statistical Analysis*

Since no major differences were observed between different sowing dates on the plants' physiology and phenology, the sowing dates were combined together in a two-year experimentation (2017–2018 and 2018–2019). For this reason, sowing-date and year were not considered as independent factors, whereas irrigation and N-fertilization were the only independent factors of statistical interest. The analysis of variance (ANOVA) and Tukey's HSD post-hoc test were used to estimate the mean variation among and within groups for different crop parameters, as well as to examine the effect of irrigation and N-fertilization and its interaction on a wide-range of crop parameters. The ANOVA and Tukey's HSD post-hoc test was done using Minitab 18 and IBM SPSS software. Finally, the Pearson correlation coefficient (*r*) was used to test the association or correlation between two variables.

## 3. Results

The two-year experimental field was characterized for being sandy–loam in the first layer (0–20 cm) and sandy–clay–loam in the second layer (20–40 cm), besides having slightly acidic properties (Table 1). Mineral nitrogen (in the form of ammonium-$NH_4$ and nitrate-$NO_3$) in the soil was very low with N-content lower than 0.036% at all depths (0–20 cm and 20–40 cm) and for the two years. The organic matter found in the second year of experimentation had higher values (0.60%) when compared to that of the first year (0.48%). The ratio of C/N was higher for the second year of experimentation (9.8) than for the first year (8.8). The P-availability within the first soil layer was considerably higher for the second year-experiment (44 mg kg$^{-1}$) when comparing to that of the first year (4 mg kg$^{-1}$); while similar K-availability (between 79 and 90 mg kg$^{-1}$) was found for both years.

The mean-maximum temperatures (Figure 2) during the growing cycle for different sowing dates were the following: 34.8 °C (25-Oct.), 34.6 °C (4-Nov.), 35.0 °C (19-Nov.), and 34.8 °C (8-Dec.). Moreover, at quinoa's most sensitive phenological phase, flowering (at 40 DAS and lasting for 10 days), the mean-maximum temperatures for each of the sowing dates was: 34.6 °C (25-Oct.), 33.5 °C (4-Nov.), 33.3 °C (19-Nov.), and 34.8 °C (8-Dec.). For the net irrigation requirements (Table 2), the amount of water supplied under FI was between 394 mm (8-Dec.) and 457 mm (25-Oct. and 19-Nov.); whereas for PD between 319 mm (8-Dec.) and 328 mm (4-Nov.). On the other side, under DI and EDI, the amount of irrigation applied was between 211 mm (25-Oct.) and 246 mm (4-Nov.).

**Table 1.** Soil main physico-chemical characteristics of the two-year experimentation (2017–2018 and 2018–2019) (average of 5 samples).

| Parameter | Depth/Unit | 2017–2018 | | 2018–2019 | |
|---|---|---|---|---|---|
| | | 0–20 cm | 20–40 cm | 0–20 cm | 20–40 cm |
| Sand | % | 67.2 | 54.6 | 75.3 | 59.5 |
| Silt | % | 17.6 | 16.5 | 14.8 | 12.7 |
| Clay | % | 15.2 | 28.9 | 9.9 | 27.8 |
| Texture | | Sandy–Loam | Sandy–Clay–Loam | Loamy–Sand | Sandy–Clay–Loam |
| pH ($H_2O$) | | 6.51 | 5.95 | 6.09 | 5.87 |
| C | % | 0.28 | 0.23 | 0.35 | 0.30 |
| Organic matter | % | 0.48 | 0.39 | 0.60 | 0.51 |
| N | % | 0.032 | 0.026 | 0.036 | 0.028 |
| C/N | | 8.8 | 8.7 | 9.8 | 10.6 |
| P available | mg kg$^{-1}$ | 4.0 | 1.70 | 44.0 | 31.3 |
| K available | mg kg$^{-1}$ | 79.73 | 74.97 | 90.3 | 116.0 |
| Bulk density | g cm$^{-3}$ | 1.61 | - | - | - |

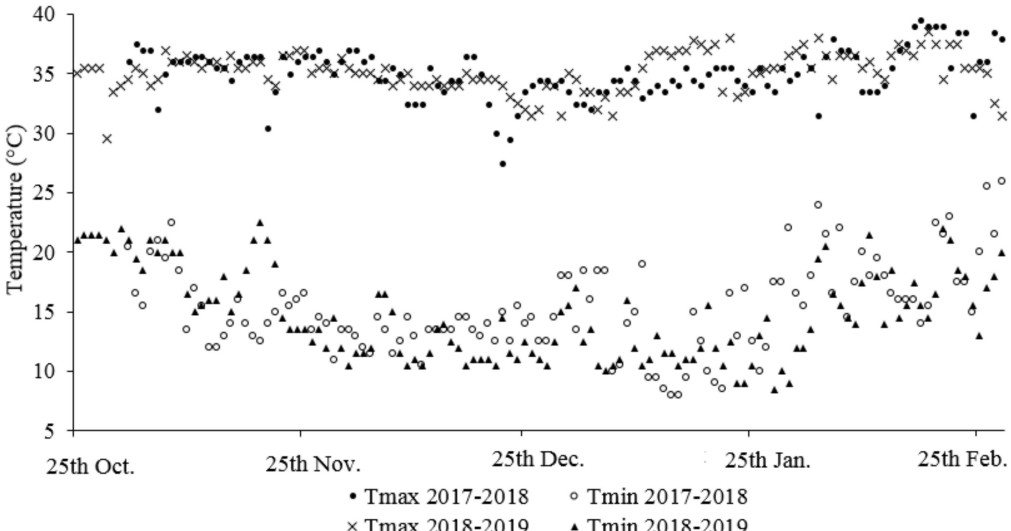

**Figure 2.** Maximum and minimum temperatures recorded during the two year-experimentation (2017–2018 and 2018–2019).

**Table 2.** Net irrigation (in mm) under full irrigation (FI), progressive drought (PD), deficit irrigation (DI), and extreme deficit irrigation (EDI) for 2017–2018 (4-Nov. and 8-Dec.) and 2018–2019 (25-Oct. and 19-Nov.).

| Irrigation Schedule | 25-Oct. | | | 4-Nov. | | | 19-Nov. | | | 8-Dec. | | |
|---|---|---|---|---|---|---|---|---|---|---|---|---|
| | Amount | Events | μ | Amount | Events | μ | Amount | Events | μ | Amount | Events | μ |
| FI | 457 | 27 | 16.9 | 410 | 38 | 10.8 | 457 | 26 | 17.6 | 394 | 28 | 14.1 |
| PD | - | - | - | 328 | 37 | 8.9 | - | - | - | 319 | 29 | 11.0 |
| DI | - | - | - | 246 | 34 | 7.2 | - | - | - | 228 | 26 | 8.8 |
| EDI | 211 | 29 | 7.3 | - | - | - | 227 | 27 | 8.4 | - | - | - |

Legend: (a) Irrigation: FI (full irrigation); PD (progressive drought); DI (deficit irrigation); EDI (extreme deficit irrigation). Note: Irrigation and rain equal to or more than 1 mm; amount (in mm); Events (in n°); μ (average water supply per irrigation event).

Different sowing dates did not have a major impact on quinoa's phenological phases (Figure 3), as mean-temperatures and photoperiod did not differ much among sowing dates (max $\Delta T$ of 0.4 °C and max $\Delta$ photoperiod of 26 min between sowing dates). For the early sowing dates (25-Oct. and 4-Nov.), higher temperatures were recorded at early-vegetative stages while lower at anthesis, and vice-versa for late sowing dates (19-Nov. and 8-Dec.). The observed photoperiodicity at this time of

the year, and for this location, was characteristic of tropical zones, with approximately 12 h of daytime and nighttime. The appearance of two-cotyledons was relatively fast, 5 DAS, being faster among experiments sown on 25-Oct. and 19-Nov. This was caused by higher diurnal temperatures (>20 °C) when sowing on the 25-Oct. and 19-Nov, than for the 8-Dec. (<15 °C). Similar growing rates were observed at two and four leaves; with changing patterns at eight leaves, with a lower rate amongst plants sown in 19-Nov. and 8-Dec. Panicle formation was totally reached at 30 DAS for all sowing dates, while flowering at 40 DAS. Nevertheless, the time for reaching flowering was slightly earlier amongst plants sown on the 25-Oct. and 4-Nov, than among those plants sown on the 19-Nov. and 8-Dec. Quinoa milky grains were observed at 60 DAS, while physiological maturity between 83 and 90 DAS.

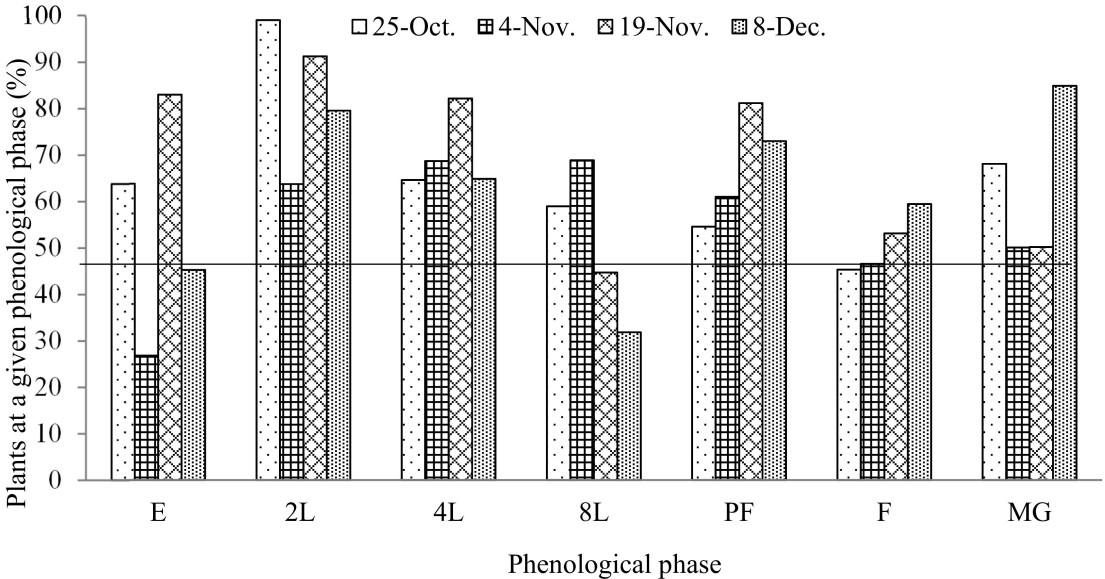

**Figure 3.** Emergence (E), two leaves (2L), four leaves (4L), eight leaves (8L), panicle formation (PF), flowering (FL), and milky grains (MG) of quinoa at 5, 15, 18, 24, 30, 40, and 60 days after sowing (DAS), respectively, for different sowing dates (25-Oct., 4-Nov., 19-Nov., and 8-Dec.).

For the cumulative growing degree days (CGDD in °C) there was a perfect negative correlation ($r$) between phenological phases and sowing dates (Table 3). In fact, there was a negative relationship between CGDD and sowing dates, with higher accumulation of degrees amongst early-sown plants (25 Oct. and 4 Nov.). In particular, the correlation coefficient was highest ($r \geq 0.79$) up to 60 DAS, while decreasing at physiological maturity. This was explained by the fact that harvesting was carried out in different dates, being earlier amongst late-sowing plants (83 DAS).

Tables 4 and 5 report the interaction between irrigation and N-fertilization (independent factors) and its effect on different crop parameters (dependent factors). Even though there was a decreasing trend on the observed parameter-values under drought-stress conditions and N-fertilization reduction, the measured crop parameters were, in many cases, not statistically significant different when N-fertilization and irrigation interacted together (Table 4); except for yield (Y) and biomass (AGB) in 2017–2018 under FI and 100 kg N ha$^{-1}$ (1171 and 3341 kg ha$^{-1}$).

Furthermore, as observed in Table 5, the highest yields and biomass (AGB) were observed amongst plants exposed to PD, with 1012 kg ha$^{-1}$ and 2436 kg ha$^{-1}$, respectively, rather than on FI plants. On the contrary, the lowest yields and biomass were observed under DI (663 and 1928 kg ha$^{-1}$, respectively) and EDI (480 and 1321 kg ha$^{-1}$, respectively). Statistically significant differences were reported when comparing the yields of PD and EDI. For N-fertilization, the highest yields and biomass were observed when applying 25 kg N ha$^{-1}$ (1038 and 2426 kg ha$^{-1}$, respectively); hence, increasing N-fertilization did not result in a higher crop development nor yield. Statistically significant differences were reported

for the weight of thousand grains (TGW), which were lower under EDI and 0 kg N ha$^{-1}$ (1.37 g). The harvest index (HI) remained constant for all irrigation levels (around 37% for FI, PD and EDI), except for EDI with a value of 29%. For the crop water productivity (CWP), the mean value of all irrigation levels (FI, PD, DI, and EDI) was 0.493 kg m$^{-3}$, having a 2:1 ratio of biomass production per m$^3$ of water applied. Nevertheless, a higher CWP was reported for PD and DI plants, 0.576 and 0.683 kg m$^{-3}$, respectively. This shows that quinoa was capable of producing the same or more biomass with less water inputs, therefore using water more efficiently. This was corroborated when comparing PD and DI with FI, the latter having a CWP value of 0.373 kg m$^{-3}$.

Furthermore, deeper roots (RD) were observed among EDI (11.9 cm), with significant differences ($p < 0.05$) when comparing to PD (6.5 cm) and DI (7.5 cm). The opposite situation was found for the root horizontal length (RHL), being wider among FI plants (19.8 cm), than DI (11.2 cm) and EDI (12.3 cm). The surface water coverage of FI plots was greater, hence roots expanded side-ways. For the branching (B) and stem diameter (ST), plots receiving 25 kg N ha$^{-1}$ had higher values, with 10 branches per plant and 0.65 cm stem-diameter. Overall, there was a positive relationship between plant height and irrigation but was not stronger enough to be considered as statistically different. The highest plants were observed under FI and PD (40.2 and 40.9 cm), whereas the smallest under EDI (34.5 cm). For N-fertilization, the highest plants were also reported among plots supplied with 25 kg N ha$^{-1}$ (43.2 cm) and the smallest under 0 kg N ha$^{-1}$ (36.7 cm).

**Table 3.** Cumulative growing degree days (CGDD in °C) for different phenological stages and photoperiodicity (minutes per day) for different sowing dates (25-Oct., 4-Nov., 19-Nov. and 8-Dec.).

| Phenological Phase | 25-Oct. | 4-Nov. | 19-Nov. | 8-Dec. | Pearson (CGDD versus Sowing Date) |
|---|---|---|---|---|---|
| Emergence (°C) | 123.8 | 123.8 | 119.0 | 101.0 | −0.93 |
| 2 Leaves (°C) | 370.3 | 343.5 | 329.9 | 307.0 | −0.98 |
| 4 Leaves (°C) | 438.8 | 405.5 | 395.9 | 366.3 | −0.97 |
| 8 Leaves (°C) | 578.3 | 542.5 | 515.7 | 495.8 | −0.97 |
| Panicle formation (°C) | 722.8 | 671.5 | 637.4 | 624.5 | −0.92 |
| Flowering (°C) | 933.7 | 877.3 | 831.2 | 821.5 | −0.92 |
| Milky grains (°C) | 1340.7 | 1293.8 | 1241.1 | 1263.0 | −0.79 |
| Maturity (°C) | 1866.7 | 1919.8 | 1785.5 | 1832.3 | −0.55 |
| Harvest (DAS) | 86 | 90 | 85 | 83 | - |
| Mean photoperiod (min day$^{-1}$) | 692.9 | 692.6 | 692.7 | 696.3 | - |
| Min. photoperiod (min day$^{-1}$) | 688.1 | 688.1 | 688.1 | 688.1 | - |
| Max. photoperiod (min day$^{-1}$) | 707.3 | 701.9 | 704.5 | 714.3 | - |

**Table 4.** Post-hoc Tukey' HSD pairwise comparison test for different crop parameters and interaction among factors of study (irrigation and fertilization), mean-values for each year experimentation (2017–2018 and 2018–2019).

| I | F | PH (cm) | B (n°) | PS (cm) | SD (cm) | RD (cm) | RHL (cm) | Y (kg ha⁻¹) | AGB (kg ha⁻¹) | TGW (g) | HI (%) | CWP (kg m⁻³) |
|---|---|---|---|---|---|---|---|---|---|---|---|---|
| | | | | | | | **2017–2018** | | | | | |
| 100 | 100 | 36.4 ± 4.2 a | 7.8 ± 4.0 a | 13.8 ± 1.9 a | 0.57 ± 0.30 a | 7.8 ± 1.7 a | 20.4 ± 14.8 a,b | 430 ± 201 a | 1527 ± 439 a | 1.72 ± 0.17 a | 35 ± 4 a | 0.258 ± 0.128 b |
| 100 | 50 | 39.0 ± 5.4 a | 9.4 ± 4.5 a | 16.0 ± 3.0 a | 0.58 ± 0.11 a | 4.5 ± 2.4 a | 22.9 ± 9.4 a | 735 ± 339 a | 2309 ± 811 a | 1.76 ± 0.13 a | 40 ± 2 a | 0.428 ± 0.097 a,b |
| 100 | 25 | 37.3 ± 6.7 a | 10.4 ± 2.8 a | 15.0 ± 1.6 a | 0.66 ± 0.17 a | 7.0 ± 4.7 a | 16.8 ± 7.2 a,b | 727 ± 250 a | 1743 ± 611 a | 1.92 ± 0.20 a | 41 ± 1 a | 0.368 ± 0.128 a,b |
| 80 | 100 | 32.2 ± 11.1 a | 6.2 ± 3.7 a | 12.3 ± 2.4 a | 0.50 ± 0.12 a | 4.2 ± 1.1 a | 9.5 ± 4.4 a,b | 462 ± 175 a | 1353 ± 530 a | 1.88 ± 0.23 a | 35 ± 5 a | 0.315 ± 0.259 b |
| 80 | 50 | 44.9 ± 15.9 a | 8.9 ± 2.5 a | 16.0 ± 4.8 a | 0.59 ± 0.13 a | 8.4 ± 3.9 a | 16.6 ± 7.0 a,b | 1110 ± 620 a | 2795 ± 1675 a | 1.87 ± 0.09 a | 41 ± 5 a | 0.744 ± 0.557 a,b |
| 80 | 25 | 45.7 ± 9.9 a | 11.3 ± 4.3 a | 16.1 ± 5.5 a | 0.70 ± 0.17 a | 7.2 ± 4.0 a | 19.2 ± 5.4 a,b | 1356 ± 631 a | 2859 ± 1339 a | 1.94 ± 0.16 a | 39 ± 9 a | 0.670 ± 0.244 a,b |
| 60 | 100 | 30.6 ±10.1 a | 4.1 ± 3.0 a | 12.4 ± 3.4 a | 0.47 ± 0.11 a | 8.0 ± 4.2 a | 7.1 ± 4.9 b | 233 ± 203 a | 1370 ± 273 a | 1.89 ± 0.28 a | 31 ± 5 a | 0.259 ± 0.093 b |
| 60 | 50 | 38.9 ± 8.8 a | 5.6 ± 3.1 a | 16.2 ± 4.2 a | 0.65 ± 0.19 a | 6.5 ± 2.8 a | 11.5 ± 2.9 a,b | 588 ± 313 a | 1352 ± 635 a | 2.04 ± 0.45 a | 38 ± 10 a | 0.625 ± 0.257 a,b |
| 60 | 25 | 46.5 ± 14.7 a | 9.2 ± 4.4 a | 17.5 ± 5.0 a | 0.57 ± 0.17 a | 8.0 ± 4.3 a | 14.9 ± 3.1 a,b | 1084 ± 972 a | 2727 ± 1821 a | 2.16 ± 0.31 a | 40 ± 9 a | 1.095 ± 0.773 a |
| | μ | 39.0 ± 9.7 | 8.1 ± 3.6 | 15.0 ± 3.5 | 0.59 ± 0.16 | 6.8 ± 3.2 | 15.4 ± 6.6 | 747 ± 411 | 2004 ± 904 | 1.91 ± 0.23 | 38 ± 6 | 0.529 ± 0.282 |
| **I** | **F** | | | | | | | **2018–2019** | | | | |
| 100 | 100 | 46.5 ± 8.3 a | 8.4 ± 1.1 a | 16.8 ± 4.6 a | 0.71 ± 0.12 a | 14.5 ± 2.2 a | 23.4 ± 4.9 a | 1171 ± 362 a | 3341 ± 970 a | 1.65 ± 0.31 a | 30 ± 9 a | 0.468 ± 0.249 a |
| 100 | 0 | 39.8 ± 7.0 a | 7.1 ± 0.7 a,b | 14.2 ± 2.9 a | 0.59 ± 0.10 a,b | 9.3 ± 1.9 b | 15.4 ± 7.8 b | 805 ± 114 a,b | 2166 ± 748 b | 1.54 ± 0.40 a | 33 ± 10 a | 0.325 ± 0.122 a |
| 50 | 100 | 35.4 ± 10.8 a | 7.4 ± 1.5 a | 12.4 ± 3.3 a | 0.59 ± 0.11 a,b | 13.2 ± 3.2 a,b | 13.6 ± 3.1 b | 441 ± 212 b | 1293 ± 551 b | 1.40 ± 0.31 a | 28 ± 16 a | 0.357 ± 0.217 a |
| 50 | 0 | 33.7 ± 8.7 a | 5.4 ± 1.5 b | 12.1 ± 3.5 a | 0.46 ± 0.14 b | 10.7 ± 4.4 a,b | 11.0 ± 4.2 b | 519 ± 221 b | 1077 ± 321 b | 1.37 ± 0.16 a | 30 ± 12 a | 0.326 ± 0.239 a |
| | μ | 38.8 ± 8.7 | 7.1 ± 1.2 | 13.9 ± 3.6 | 0.59 ± 0.12 | 11.9 ± 2.9 | 15.9 ± 5.0 | 678 ± 189 | 2031 ± 603 | 1.49 ± 0.30 | 0.30 ± 0.12 | 0.37 ± 0.21 |

Legend: (I) Irrigation: 100% PET-Full irrigation-FI; 80% PET-Progressive drought-PD; 60% PET-deficit irrigation-DI; 50% PET-extreme deficit irrigation-EDI; (F) Fertilization: 100, 50, 25, 0 kg N ha⁻¹. Note: PH (plant height), B (n° of branches per plant), PS (length of the panicle), SD (stem diameter), RD (root depth), RL (root horizontal length), Y (yield), AGB (aboveground biomass), TGW (thousand-grain weight), HI (harvest index), CWP (crop water productivity). Note: means that do not share a letter are significantly different, $p \leq 0.05$ according to the test of Tukey's HSD. Note: Average value ± standard deviation values are shown, with three repetitions (2017–2018, average 25-Oct and 19-Nov) and four repetitions (2018–2019, average 4-Nov. and 8-Dec.).

**Table 5.** Post-hoc Tukey's HSD pairwise comparison test for different crop parameters and factors of study (irrigation and fertilization), mean of all sowing dates (25-Oct., 4-Nov., 19-Nov., and 8-Dec.).

| Irrigation | PH (cm) | B (n°) | PS (cm) | SD (cm) | RD (cm) | RHL (cm) | Y (kg ha⁻¹) | AGB (kg ha⁻¹) | TGW (g) | HI (%) | CWP (kg m⁻³) |
|---|---|---|---|---|---|---|---|---|---|---|---|
| 100 | 40.2 ± 7.6 a | 8.5 ± 3.1 a | 15.2 ± 3.3 a | 0.63 ± 0.18 a | 9.0 ± 4.4 a,b | 19.8 ± 9.7 a | 727 ± 281 a | 2321 ± 998 a | 1.71 ± 0.30 b | 36 ± 8 a | 0.373 ± 0.176 b |
| 80 | 40.9 ± 14.0 a | 8.8 ± 4.1 a | 14.8 ± 4.8 a | 0.60 ± 0.17 a | 6.5 ± 3.7 b | 15.0 ± 7.0 b | 1012 ± 648 a | 2436 ± 1468 a | 1.90 ± 0.17 a,b | 38 ± 7 a | 0.576 ± 0.425 a |
| 60 | 38.6 ± 13.2 a | 6.3 ± 4.1 b | 15.4 ± 4.8 a | 0.56 ± 0.18 a | 7.5 ± 3.9 b | 11.2 ± 4.9 b | 663 ± 724 a | 1928 ± 1421 a,b | 2.03 ± 0.37 a | 37 ± 9 a | 0.683 ± 0.592 a |
| 50 | 34.5 ± 9.9 a | 6.4 ± 1.8 b | 12.2 ± 3.4 a | 0.53 ± 0.14 a | 11.9 ± 4.0 a | 12.3 ± 4.0 b | 480 ± 220 b | 1321 ± 397 b | 1.38 ± 0.25 c | 29 ± 15 a | 0.340 ± 0.230 b |
| **Fertilization** | **PH (cm)** | **B (n°)** | **PS (cm)** | **SD (cm)** | **RD (cm)** | **RHL (cm)** | **Y (kg ha⁻¹)** | **AGB (kg ha⁻¹)** | **TGW (g)** | **HI (%)** | **CWP (kg m⁻³)** |
| 100 | 36.8 ± 11.0 a | 6.9 ± 3.1 b,c | 13.6 ± 3.8 a | 0.58 ± 0.18 b | 10.0 ± 4.7 a | 15.3 ± 9.6 a,b | 483 ± 302 b | 2032 ± 1116 a | 1.71 ± 0.32 b | 32 ± 11 b | 0.343 ± 0.222 b |
| 50 | 40.9 ± 11.3 a | 7.9 ± 3.8 b | 16.1 ± 4.1 a | 0.60 ± 0.15 a,b | 6.3 ± 3.4 b | 17.0 ± 8.5 a | 806 ± 489 a,b | 2141 ± 1310 a | 1.89 ± 0.30 a | 40 ± 7 a | 0.599 ± 0.381 a,b |
| 25 | 43.2 ± 11.7 a | 10.3 ± 4.0 a | 16.2 ± 4.5 a | 0.65 ± 0.18 a | 7.4 ± 4.4 a,b | 17.0 ± 5.8 a | 1038 ± 733 a | 2426 ± 1413 a | 2.01 ± 0.26 a | 40 ± 8 a | 0.711 ± 0.560 a |
| 0 | 36.7 ± 8.5 a | 6.2 ± 1.4 c | 13.1 ± 3.4 a | 0.52 ± 0.14 b | 10.0 ± 3.4 a | 13.2 ± 6.7 b | 673 ± 223 a,b | 1789 ± 768 a | 1.45 ± 0.32 c | 32 ± 11 b | 0.326 ± 0.190 b |

Legend: Irrigation: 100% PET-Full irrigation-FI, 80% PET-Progressive drought-PD, 60% PET-deficit irrigation-DI, 50% PET-extreme deficit irrigation-EDI; Fertilization: 100, 50, 25, 0 kg N ha⁻¹. Note: PH (plant height), B (n° of branches per plant), PS (length of the panicle), SD (stem diameter), RD (root depth), RL (root horizontal length), Y (yield), AGB (aboveground biomass), TGW (thousand-grain weight), HI (harvest index), CWP (crop water productivity). Note: Means that do not share a letter are significantly different, $p \leq 0.05$ according to the test of Tukey's HSD. Note: average value ± standard deviation values are shown.

In this study, the canopy cover played an important role on light interception and plant growth, just like on the resulting yield and biomass (Table 6). For the canopy cover, statistically significant differences ($p < 0.05$) were observed between highly irrigated/fertilized treatments (FI-100 kg N ha$^{-1}$) and less irrigated/fertilized treatments (EDI-0 kg N ha$^{-1}$), both for 25-Oct. and 19-Nov. For the yield, on the 25-Oct., significant differences ($p < 0.05$) were observed when comparing FI-100 kg N ha$^{-1}$ to that of FI-0 kg N ha$^{-1}$, and the previous with EDI-100 kg N ha$^{-1}$ and EDI-0 kg N ha$^{-1}$. For 19-Nov, despite of the decreasing yields with lower N-inputs and water-application, no significant differences were reported between treatments and the interaction amongst them. For both sowing dates (25-Oct. and 19-Nov.), there was a moderate correlation ($r \geq 0.5$) between dependent variables (canopy cover, yield, and biomass) and independent variables (irrigation and N-fertilization). Overall, when associating dependent variables (parameter of study) and independent variables (controlled parameter) the Pearson correlation coefficient was higher for irrigation than for N-fertilization. For instance, when correlating biomass and yield with irrigation, the *r* value was of 0.66 and 0.85 for 25-Oct., and of 0.68 and 0.42 for 19-Nov.

**Table 6.** Relationship between different levels of irrigation and N-fertilization with the maximum canopy cover (CC as a percentage), yield (Y), and aboveground biomass (ABG) (kg ha$^{-1}$) in the second year of experimentation (25-Oct. and 19-Nov.).

| | | 25-Oct. | | | 19-Nov. | | |
|---|---|---|---|---|---|---|---|
| **I** | **F** | **Max. CC (%)** | **Y (kg ha$^{-1}$)** | **ABG (kg ha$^{-1}$)** | **Max. CC (%)** | **Y (kg ha$^{-1}$)** | **ABG (kg ha$^{-1}$)** |
| 100 | 100 | 26.6 ± 8.3 a | 1380 ± 251 a | 3522 ± 1231 a | 44.5 ± 8.9 a | 752 ± 62 a | 3205 ± 683 a |
| 100 | 0 | 13.4 ± 1.5 b | 875 ± 99 b | 2005 ± 146 b | 28.6 ± 9.8 b | 711 ± 44 a | 2326 ± 1022 a,b |
| 50 | 100 | 7.7 ± 6.6 b | 373 ± 185 c | 1280 ± 702 b | 26.4 ± 5.6 b | 508 ± 215 a | 1311 ± 224 b |
| 50 | 0 | 4.4 ± 5.3 b | 322 ± 116 c | 973 ± 336 b | 16.5 ± 6.8 b | 717 ± 80 a | 1283 ± 137 b |

Legend: (I) Irrigation: 100% PET-Full irrigation-FI, 80% PET-Progressive drought-PD, 60% PET-deficit irrigation-DI, 50% PET-extreme deficit irrigation-EDI; (F) Fertilization: 100, 50, 25, 0 kg N ha$^{-1}$. Note: Means that do not share a letter were significantly different, $p \leq 0.05$ according to the test of Tukey's HSD.

Furthermore, remarkable information was observed when analyzing crop's responses to FI and EDI (100% and 50% PET) with different levels of N-fertilization (100 and 0 kg N ha$^{-1}$) (Figure 4). Despite of the greater canopy cover expansion observed in 19-Nov. (maximum CC of 45%) when compared to 25-Oct. (maximum CC of 27%), yields were greater in 25-Oct. than in 19-Nov. under FI, and vice versa for EDI (Table 6). A greater canopy expansion on 19-Nov. when comparing to 25-Oct. was probably due to the milder temperatures during the vegetative phase. The maximum canopy cover was observed at 50 DAS (25-Oct.) and at 60 DAS (19-Nov.), with a respective soil percentage covered of 27% and 45% for FI-100 kg N ha$^{-1}$, and of 4% and 17% for EDI-0 kg N ha$^{-1}$. In Table 7, the effect of irrigation on canopy cover was statistically significantly different ($p < 0.05$) for most of the observations and sowing dates, having a larger canopy cover those plants that were highly irrigated (FI). A Similar statistical difference trend ($p < 0.05$) was observed between higher and lower levels of N-fertilization; being much greater than the canopy expansion amongst highly fertilized plants. In particular, these differences were depicted from observation N°4 onwards, when N-fertilization had already been assimilated by plants. In Table 8, for 25-Oct. observation N°5, the interacting effect of higher irrigation with nitrogen fertilization on the canopy cover was clear, showing statistical differences ($p < 0.05$) among most groups of means. Despite of the decreasing canopy cover with lower irrigation and N-fertilization, significant differences were only observed for the FI-100 kg N ha$^{-1}$ when compared to the rest of treatments (FI-0 kg N ha$^{-1}$, EDI-100 kg N ha$^{-1}$, and EDI-0 kg N ha$^{-1}$).

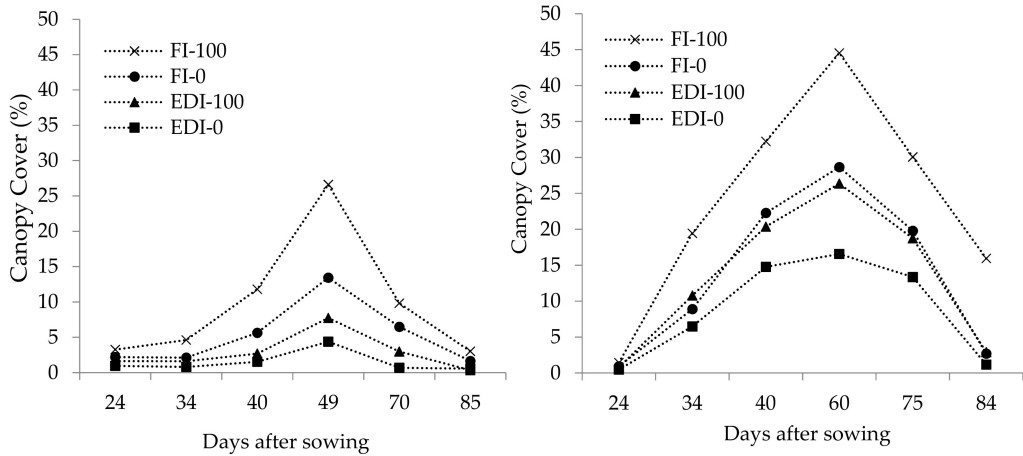

**Figure 4.** Canopy cover during the second year of experimentation, **left** (25-Oct.) and **right** (19-Nov.). Note: FI-100 (full irrigation-FI and 100 kg N ha$^{-1}$); FI-0 (full irrigation-FI and 0 kg N ha$^{-1}$); EDI-100 (extreme deficit irrigation-EDI and 100 kg N ha$^{-1}$); EDI-0 (extreme deficit irrigation-EDI and 0 kg N ha$^{-1}$).

**Table 7.** Post-hoc Tukey's HSD pairwise comparison test for the canopy cover (%) during the second year of experimentation.

| | N°1 | N°2 | N°3 | N°4 | N°5 | N°6 |
|---|---|---|---|---|---|---|
| **25-October** | | | | | | |
| **I** | **N°1** | **N°2** | **N°3** | **N°4** | **N°5** | **N°6** |
| 100 | 2.7 ± 1.3 a | 3.4 ± 3.6 a | 8.7 ± 5.7 a | 20.0 ± 9.5 a | 8.2 ± 3.2 a | 2.3 ± 1.4 a |
| 50 | 1.3 ± 0.5 b | 1.2 ± 0.5 a | 2.1 ± 1.6 b | 6.1 ± 6.6 b | 0.7 ± 0.4 b | 0.5 ± 0.5 b |
| **F** | | | | | | |
| 100 | 2.4 ± 1.1 a | 3.1 ± 3.5 a | 7.2 ± 6.6 a | 17.2 ± 12.9 a | 5.3 ± 5.2 a | 1.7 ± 1.8 a |
| 0 | 1.6 ± 1.2 a | 1.5 ± 1.3 a | 3.6 ± 3.0 a | 8.9 ± 6.4 b | 3.6 ± 3.6 a | 1.1 ± 0.9 a |
| **19-November** | | | | | | |
| **I** | **N°1** | **N°2** | **N°3** | **N°4** | **N°5** | **N°6** |
| 100 | 1.2 ± 0.6 a | 14.2 ± 8.0 a | 27.3 ± 10.7 a | 36.6 ± 13.1 a | 24.9 ± 7.7 a | 9.3 ± 9.0 a |
| 0 | 0.6 ± 0.5 b | 8.6 ± 5.4 a | 17.6 ± 9.3 a | 21.5 ± 8.0 b | 16.0 ± 4.7 b | 2.0 ± 1.3 b |
| **F** | | | | | | |
| 100 | 1.1 ± 0.7 a | 15.1 ± 8.2 a | 26.3 ± 12.4 a | 35.4 ± 12.5 a | 24.4 ± 8.1 a | 9.4 ± 8.8 a |
| 0 | 0.7 ± 0.4 a | 7.7 ± 3.5 b | 18.5 ± 8.1 a | 22.6 ± 10.7 b | 16.6 ± 5.0 b | 1.9 ± 1.6 b |

Legend: (I) Irrigation: 100% PET-Full irrigation-FI and 50% PET-extreme deficit irrigation-EDI; (F) Fertilization: 100 and 0 kg N ha$^{-1}$. Note: N°1 to N°6 correspond to measurement dates: 24, 34, 40, 49, 70 and 85 DAS (25-Oct.); and to 24, 34, 40, 60, 75 and 84 DAS (19-Nov.). Note: means that do not share a letter were significantly different, $p \leq 0.05$ according to the test of Tukey-HSD.

**Table 8.** ANOVA test for the canopy cover (%) and its interaction between irrigation and fertilizer application.

| I | F | N°1 | N°2 | N°3 | N°4 | N°5 | N°6 |
|---|---|---|---|---|---|---|---|
| **25-Oct.** | | | | | | | |
| 100 | 100 | 3.3 ± 0.8 a | 4.6 ± 4.8 a | 11.8 ± 6.6 a | 26.6 ± 9.5 a | 9.8 ± 2.7 a | 3.0 ± 1.5 a |
| 100 | 0 | 2.2 ± 1.5 a,b | 2.1 ± 1.7 a | 5.6 ± 2.7 b | 13.4 ± 1.7 b | 6.5 ± 2.9 b | 1.6 ± 0.8 a,b |
| 50 | 100 | 1.6 ± 0.5 b | 1.6 ± 0.4 a | 2.7 ± 1.7 b | 7.7 ± 7.6 b | 0.7 ± 0.3 c | 0.3 ± 0.1 b |
| 50 | 0 | 1.0 ± 0.3 b | 0.8 ± 0.3 a | 1.5 ± 1.4 b | 4.4 ± 6.1 b | 0.7 ± 0.5 c | 0.6 ± 0.7 b |
| **19-Nov.** | | | | | | | |
| **I** | **F** | **N°1** | **N°2** | **N°3** | **N°4** | **N°5** | **N°6** |
| 100 | 100 | 1.4 ± 0.6 a | 19.4 ± 8.1 a | 32.2 ± 11.8 a | 44.5 ± 10.3 a | 30.1 ± 7.4 a | 15.9 ± 8.3 a |
| 100 | 0 | 0.9 ± 0.4 a,b | 8.9 ± 3.3 b | 22.3 ± 7.9 a,b | 28.6 ± 11.3 b | 19.8 ± 3.5 b | 2.9 ± 1.9 b |
| 50 | 100 | 0.8 ± 0.6 a,b | 10.8 ± 6.5 a,b | 20.4 ± 11.2 a,b | 26.4 ± 6.5 b | 18.7 ± 3.6 b | 2.7 ± 1.1 b |
| 50 | 0 | 0.5 ± 0.3 b | 6.5 ± 3.6 b | 14.8 ± 7.4 b | 16.5 ± 6.7 b | 13.4 ± 4.4 b | 1.2 ± 0.9 b |

Legend: (I) Irrigation: 100% PET-Full irrigation-FI and 50% PET-extreme deficit irrigation-EDI; (F) Fertilization: 100 and 0 kg N ha$^{-1}$. Note: N°1 to N°6 correspond to measurement dates: 24, 34, 40, 49, 70 and 85 DAS (25-Oct.); and to 24, 34, 40, 60, 75 and 84 DAS (19-Nov.). Note: means that do not share a letter were significantly different, $p \leq 0.05$ according to the test of Tukey's HSD.

## 4. Discussion

Mean-temperatures recorded during the vegetative stage (27.5 °C for 25-Oct. and 25.5 °C for 19-Nov. average of temperature 40 DAS) were close to quinoa's optimal growth temperatures, 10–25 °C [29]. A closer temperature to optimal growth for the sowing on the 19-Nov resulted in greater canopy expansion (47%), when compared to 25-Oct (27%). In addition, this research was conducted in a much warmer climate (25–30 °C) to that of its origin (15–20 °C) Bolivian Altiplano [29]. This research findings differ to those of Gifford et al. [30], showing a tight relationship between light interception and production of grains and biomass. In fact, maximum temperatures occurring at anthesis were critical for quinoa pollination, resulting in the reduction of pollen production and viability [14,20]. Therefore, despite the greater canopy expansion observed in 19-Nov, the yields were lower (752 kg ha$^{-1}$) to those reported on the 25-Oct (1380 kg ha$^{-1}$). This research also confirms that quinoa can stand higher temperature thresholds at anthesis (38 °C) than other cereals grown in Burkina Faso; for instance, rice 35 °C, maize 35 °C or sorghum 34 °C [31–33]. Apart from temperature, photoperiodicity plays an important role on the rate of plant growth. In fact, c.v. *Titicaca* had a higher photoperiod-sensitivity in comparison to other quinoa varieties which were not affected by changes in photoperiodicity [34]. For tropical latitudes (11° N, Burkina Faso), the length of the growing cycle of c.v. *Titicaca* was much shorter, 83–90 days, to that observed in subtropical regions, 169 days and 134 days for maturity, respectively, for Iraq (32° N) and Germany (44° N); but having a similar growing cycle to that of Yemen (14° N), 118 days [35–37]. However, in this research, different sowing dates, with slightly different photoperiods (max Δ photoperiod of 26 min between sowing dates), did not have an impact on the duration of the growing cycle. In respect to the growing degree days (GDDs), having 3 °C as a base temperature; very similar values to this research were reported by Präger et al. [37] for c.v. *Titicaca* in Germany for the year 2015. Based on their results, the cumulative GDD for c.v. *Titicaca* was 1874 °C; whereas for Burkina Faso, the mean cumulative GDD for all sowing dates was 1851 °C.

Other abiotic factors, such as irrigation and N-fertilization, were of interest within this study. For irrigation, there was an increase in CWP with decreasing irrigation; except for EDI, that resulted in crop failure due to the excessive water stress. Simulated conditions for quinoa showed CWP values of 0.3 to 0.6 kg m$^{-3}$ with crop evapotranspiration varying between 200 mm to 400 mm [21]. These values were in accordance with those reported in Burkina Faso with a CWP value of 0.57 kg m$^{-3}$ for 325 mm (PD). In fact, in this research, a similar curve was observed to that of Geerts and Raes [21], with higher CWP between 250–400 mm, while having lower CWP values above or below the previous evapotranspiration-thresholds. The results of this research were in harmony with those observed under field conditions in Bolivia (2005–2006), with CWP values of 0.38 to 0.45 kg m$^{-3}$ under FI (more than 500 mm) and DI (around 400 mm), respectively [38]. Overall, even if quinoa could stand water-applications as low as 200 mm, the crop sacrificed most of its yield for survival (480 kg ha$^{-1}$ under EDI compared to 1012 kg ha$^{-1}$ under PD, average of all sowing dates). So far, in the literature, only one study has examined such water stressors (183 mm during the growing season) to those of this research (211 mm in 85 days growing cycle, equal to 2.48 mm day$^{-1}$); but with much lower evapotranspiration to that of Burkina Faso [39]. In addition, emerging findings corroborate quinoa's high-water use efficiency under drought-stress conditions and were in harmony with those results reported in the literature [14,19,38,40]. In fact, rapid stomata closure, sunken stomata, restricted shoot growth, accelerated leaf-senescence were among quinoa's physiological responses to drought-stress conditions, while given the plant an optimal adaptability to dry environments. For nitrogen fertilization, little information emerged from this study. Our findings fall within those in the literature acknowledging that N-fertilization plays a role in quinoa growth up to 25 kg N ha$^{-1}$ [24]. Between 0 to 25 kg N ha$^{-1}$ significant differences were observed for different crop parameters (plant height, yield, biomass, thousand-grain weight, among others), but no differences were depicted at higher N-applications (50 and 100 kg N ha$^{-1}$); therefore N-stabilization was reached at 25 kg N ha$^{-1}$. On the contrary, many other studies have pointed out that quinoa yields and biomass increase with higher N-fertilization, but stabilize at 80 kg N ha$^{-1}$ [22,41]. Overall, the previous findings on abiotic

factors support Atkinson and Porter's [42] insights on faster development among species growing in environments with higher temperatures and water stress conditions.

Regarding physiological parameters, the 1000-seed weight (kernel) observed in this research (2.03 g under PD and 1.38 g for EDI) was lower to that reported in other field studies in Turkey (2.1–3.2 g) and Bolivia (3–6 g) [43,44]. Low seed weight in Burkina Faso was the result of high temperatures and longer photoperiods. For cereals, a 3.1 day shortening of the grain filling phase has been fixed per degree Celsius increase in mean daily temperatures, as well as a decrease of 2.8 mg kernel per degree Celsius increase [45]. This could explain, to some extent, the differences in observed kernel weight between Burkina Faso and elsewhere. Moreover, the observed shallow root system (average 8.7 cm) was the result of high bulk density (1.61 g cm$^{-3}$). In this case, soil compaction was a limiting factor for the root-system to uptake water and nitrogen from deeper zones. This was corroborated by Daddow and Warrington [46], pointing out that bulk densities higher than 1.59 g cm$^{-3}$ for sandy-loam soils was a restrictive factor for root penetration. Poor root colonization in Burkina Faso contrasts to that observed in Bolivia and Chile, where quinoa roots have exceeded 50 cm depth [47].

Finally, observed c.v. *Titicaca* yields under FI-100 kg N ha$^{-1}$ (1380 kg ha$^{-1}$ for 25 Oct.) and PD-25 kg N ha$^{-1}$ (1356 kg ha$^{-1}$, average of 2017–2018 experiment) were higher to those observed in Egypt (1034 kg ha$^{-1}$ under 90 kg N ha$^{-1}$), in line with those reported in Iraq (1270 kg ha$^{-1}$), and lower to those observed in Germany (1980 kg ha$^{-1}$) and Yemen (2100 kg ha$^{-1}$) [35–37,41].

## 5. Conclusions

The length of the growing cycle for c.v. *Titicaca* in Burkina Faso has shown a photoperiod-sensitivity when compared to other regions in the world. Growing quinoa in Burkina Faso when temperatures are closer to the optimal temperatures for growth can increase the biomass but does not necessarily determine the final yield. In fact, the yield is tightly dependent on the heat stress and water vapor pressure deficits occurring at flowering; if temperatures break a given temperature threshold—around 38 °C—the plants can become sterile. The yields have been stabilized under full irrigation and progressive drought, meaning that further water stressors—DI and EDI—will result in considerable yield losses. Under field conditions, emerging findings have shown that plant's development and growth was affected, to some extent, by N-fertilization; nevertheless, in general, quinoa N-requirements were low (25 kg N ha$^{-1}$). Moreover, high temperatures and water vapor pressure deficits during anthesis, above-optimal temperatures for growth, compacted sandy–loam soils under field conditions, and a short growing cycle are among the abiotic factors that reduce quinoa's yields in the Sahel. However, agricultural management strategies can be tailored for enhancing quinoa yields according to the needs and means of this region. For instance, a two-pass mechanical tillage incorporating organic fertilizer (as it has a longer mineralization than inorganic fertilizers) prior to the sowing is recommended. Frequent irrigation of the soil, but in small quantities in order to reduce high water percolation typically of sandy–loam soils is highly suggested for avoiding soil compaction and creating favorable conditions for the plant root system. Planning of agricultural activities, particularly through a well sowing calendar, is crucial for quinoa in the Sahel. Temperatures during the vegetative stage and heat stress at flowering have to be as close as possible to the mean optimal temperatures for growth. For this reason, this research proposes the following sowing quinoa dates for Burkina Faso: mid-November in the Soudanian agro-climatic zone, late-November in the Soudano-Sahelian zone, and early-December in the Sahelian zone.

Finally, agronomic guidelines for growing quinoa in the country need to be prepared and come in hand with an appropriate scientific knowledge-transfer towards local communities, which will then take over and be in charge of scaling-up the crop throughout the country. Further research must be tailored according to the abiotic stresses found within this region. For this reason, heat- and drought-resistant quinoa, wind-tolerant, and short-cycle varieties should be within the scope of genetic breeders.

**Author Contributions:** J.A.-B. (experimental design, data-collection and writing), A.D. (supervision and revision), A.D.M. (supervision and revision), C.S. (data-collection), P.C. (revision), J.S. (supervision) and S.O. (supervision, revision and financial support).

**Funding:** This research did not receive external funding.

**Acknowledgments:** We would like to thank the Food and Agricultural Organization (FAO) for quinoa seed provision under the Technical Cooperation Programs (TCP/SFW/3404 and TCP/RAF/3602). To Institut de l'Environnement et de Recherches Agricoles (INERA) for facilitating an experimental field during the two-year experiment (2017–2019). Special recognitions to the staff of INERA, professors, technicians, student's, workforce, and women for helping, accepting, appropriating, and taking over this research project for promoting quinoa in Burkina Faso and enhancing food security within the country.

**Conflicts of Interest:** The authors declare no conflict of interest.

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
