# Peer review of "Effect of Drought, Nitrogen Fertilization, Temperature and Photoperiodicity on Quinoa Plant Growth and Development in the Sahel"

_agronomy, doi:10.3390/agronomy9100607_

Round 1
Reviewer 1 Report
The manuscript entitled "Effect of drought, nitrogen fertilization, temperature and photoperiodicity on quinoa plant growth and development in the Sahel" reported complex data about a two-years investigation in Burkina Faso.
The aim of the work is the study of the main factors that could affect the cultivation of quinoa in Burkina Faso: drought, nitrogen, temperature and sowing date.
Even if the applicability of these data are restricted to a very limited area, the manuscript is written very well, and the experiment design is solid.
I do not have any particular suggestions.
The authors missed to fill the section "Author contribution".
Author Response
Regarding Author Contribution, this aspect has been specified during the submission process so it should appear in the final document.
Reviewer 2 Report
This is a second review.
Line 228 there is a reference to growing degree days. I did not check to see if the methodology was described elsewhere in the paper, but the readers need to know the base temperature used to calculate the growing degree days: 0 degrees or 10 degrees, or something else.
line 322 the abbreviations in the header of the table are not defined. What does Y mean (if it is yield, there is no need to abbreviate), ABG, etc. Either in footnote or in the title.
Author Response
Both comments have been included in Materials and Methods and Results (highlighted in red).
The formula of Growing Degree Days for quinoa has been added.
Tables 4 and 5, foot notes and abbreviations of crop parameters have been included.
This manuscript is a resubmission of an earlier submission. The following is a list of the peer review reports and author responses from that submission.
Round 1
Reviewer 1 Report
The manuscript "Effect of drought, nitrogen fertilization, temperature and photoperiodicity on quinoa plant growth and development in the Sahel" presents interesting data about the quinoa cultivation in Burkina Faso.
The experiment design was well explained, but the authors should explain why they used different condition between 2017/18 and 2018/19 (three different irrigation and three N fertilization in the first year, and two irrigation and two N in the second year). In this way, the authors were not able to statistically compare the two years.
Also in the rest of the paper, I found some statistical weakness.
Table 4 should be completed with data about ANOVA. Since they have two different treatments (irrigation and fertilization), a 2-way ANOVA is suggested. The interaction between the two treatments should be presented, too.
Table4: the data of the two years should be presented in different tables (or rows/columns).
Table 4: please report the statistic error for the data reported in table 4.
The authors missed to report the statistic errors also in the other tables and figures.
I suggested the support of a statistician to revise the entire paper.
Reviewer 2 Report
The fact that the two experiments were designed somewhat differently is slightly problematic and care should be taken by the authors to separate the results and discussion between the two years, accordingly. Units were generally lacking on tables. It was not clear exactly what was correlated in some areas of the paper. For example correlating with irrigation is confusion as irrigation per se is not a quantitative variable. Perhaps it was the amount of water applied that was correlated to another dependant variable?
